# Easy access to nucleophilic boron through diborane to magnesium boryl metathesis

Anne-Frédérique Pécharman[1], Annie L. Colebatch[1], Michael S. Hill[1], Claire L. McMullin[1], Mary F. Mahon[1] & Catherine Weetman[1]

Organoboranes are some of the most synthetically valuable and widely used intermediates in organic and pharmaceutical chemistry. Their synthesis, however, is limited by the behaviour of common boron starting materials as archetypal Lewis acids such that common routes to organoboranes rely on the reactivity of boron as an electrophile. While the realization of convenient sources of nucleophilic boryl anions would open up a wealth of opportunity for the development of new routes to organoboranes, the synthesis of current candidates is generally limited by a need for highly reducing reaction conditions. Here, we report a simple synthesis of a magnesium boryl through the heterolytic activation of the B–B bond of bis(pinacolato)diboron, which is achieved by treatment of an easily generated magnesium diboranate complex with 4-dimethylaminopyridine. The magnesium boryl is shown to act as an unambiguous nucleophile through its reactions with iodomethane, benzophenone and N,N′-di-isopropyl carbodiimide and by density functional theory.

[1] Department of Chemistry, University of Bath, Claverton Down, Bath BA2 7AY, UK. Correspondence and requests for materials should be addressed to M.S.H. (email: msh27@bath.ac.uk) or to C.L.M. (email: C.McMullin@bath.ac.uk).

Organoborane, boronate ester and boronic acid derivatives provide some of the most versatile and practically useful intermediates in synthetic and medicinal chemistry[1]. Although transition metal boryl chemistry and the transition metal-catalysed borylation of organic substrates have been areas of study for over half a century[2–6], the reactivity of boron is most commonly defined by its Lewis acidity and its reactivity as an electrophile. The realization of the first nucleophilic boryllithium reagents (for example, **1**, Fig. 1) by Yamashita, Nozaki and co-workers in 2006, thus, provided a long awaited landmark in organoelement synthesis[7–12]. In agreement with earlier *ab initio* calculations on model lithiated boranes[13], $LiBX_2$ (X = H, $CH_3NH_2OH$, F), such *N*-heterocyclic species have been characterized by density functional theory (DFT) as diamagnetic systems in which singlet boron is stabilised over its triplet state by some 20 kcal mol$^{-1}$ (ref. 9) This stabilization is attributed to the significant covalence of the boron to lithium interaction and the inductive influence of the electronegative nitrogen substituents. Although this latter feature also ensures that the electropositive boron centre retains a significant partial positive charge (for **1**; calculated natural population analysis charge on B = +0.072), these species behave as boron-centred nucleophiles by dint of a net polarization of the boron–lithium bond (in effect, $B^{\delta+}$–$Li^{\delta++}$) induced by the even lower effective nuclear charge of lithium (for **1**; calculated natural population analysis charge on Li = +0.755)[9,13]. Although a rich and varied chemistry has since arisen from the reactivity of compound **1** and related *N*-heterocyclic boryl anions with organic[9,14], transition metal[15–18] and main group electrophiles[14,19–25], alternative nucleophilic boron reagents remain very rare species[26–31]. Notable exceptions are Braunschweig's dimetalloborylene and borolyl anions (for example, **2** and **3**)[32,33] Bertrand and Kinjo's carbene-stabilised monovalent boron species (for example, **4**)[34–38] and a remarkable tricyanoborandiyl dianion[39–41]. Although the transient generation of $\{BH_2\}^-$ anion equivalents has also been achieved within the

coordination sphere of 1,3-bis-(2,6-diisopropyl-phenyl)imidazol-2-ylidene[42], the syntheses of all these species require strongly reducing and potentially problematic reaction conditions (for example, Li metal or $C_8K$) to achieve the formal B(I) oxidation state. The successful isolation of these compounds is also typically dependent on the high degree of kinetic stabilization provided by sterically demanding substituents directly about the boron centre. Both of these factors limit the broader synthetic utility of such anionic boron species. A more operable approach to the realization of nucleophilic boron has been derived from the quaternisation of one of the three-coordinate boron centres of a diborane(4) molecule by a neutral or anionic nucleophile (for example, **5** and **6**)[43–45]. Although the resultant adducts may act as viable surrogates for boron nucleophiles under (metal- and metal-free) catalytic conditions, these processes do not necessarily take place through the explicit generation of anionic boryl derivatives.

In this contribution we demonstrate that a similarly quaternised diboranate species, formed through the reaction of a readily accessible organomagnesium derivative with commercially available bis(pinacolato)diboron ($B_2pin_2$), allows facile and convenient access to the nucleophilic terminal $[Bpin]^-$ anion.

## Results

**Synthesis of a magnesium boryl.** We have recently reported that reaction of the silylborane $pinBSiMe_2Ph$ with the β-diketiminato magnesium *n*-butyl derivative $[HC\{(Me)CN(Dipp)\}_2Mgn\text{-}Bu]$ (**7**) provides ready access to the magnesium silyl species $[HC\{(Me)CN(Dipp)\}_2MgSiMe_2Ph]$ (**8**) through elimination of a *n*-BuBpin by-product[46]. This reaction was envisaged to ensue via a four-membered σ-bond metathesis transition state, the assembly of which was facilitated by the relative polarity of the Mg–C and Si–B bonds and the ability of boron to increase its coordination number in a β-position to the magnesium centre. The ease of this transformation led us to speculate that similar reactivity could be applied to the activation of the non-polar B–B bond of bis(pinacolato)diboron.

Treatment of compound **7** with one equivalent of $B_2pin_2$ provided a single new species (**9**) (Fig. 2). Compound **9** displayed two resonances in its $^{11}B$ NMR spectrum at δ 37.3 and 10.4 p.p.m. indicative of both three-coordinate ($sp^2$) and four-coordinate ($sp^3$) boron, respectively. These data are strongly reminiscent of the chemical shifts reported for the species formed, though not isolated, by the addition of *tert*-BuLi to $B_2pin_2$ in THF solution (δ 39.1, 6.4 p.p.m.)[47]. Crystallographic analysis confirmed compound **9** to be a magnesium complex of a [pinB-Bpin (*n*-Bu)]$^-$ anion coordinated to Mg(1) through the O(1) and O(3) centres of the two pinacolato boron moieties (Fig. 3a). In common with the comparable distances within compounds such as **5** and **6**, the B–B bond of compound **9** [1.7503(18) Å] is only marginally elongated in comparison to that reported for $B_2pin_2$ itself determined at both high [295 K; 1.716 Å] and low [120 K, 1.7041(15) Å] temperatures[48].

Optimization of the reaction to form compound **9** revealed the facile generation of an additional new species (**10**), the production of which could be maximized through performance of the reaction between compound **7** and $B_2pin_2$ in a 1:2 stoichiometry at room temperature. Compound **10**, which could also be synthesized by addition of a molar equivalent of $B_2pin_2$ to a solution of compound **9**, was identified as a β-diketiminato magnesium derivative of the unusual catenated triboron $[B_3pin_3]^-$ anion by a further X-ray diffraction analysis (Fig. 3b). The structure of this boron-containing anion bears some similarity to several neutral *catena*-triboranes, which have been recently described by Braunschweig *et al.*[49]. Whereas these

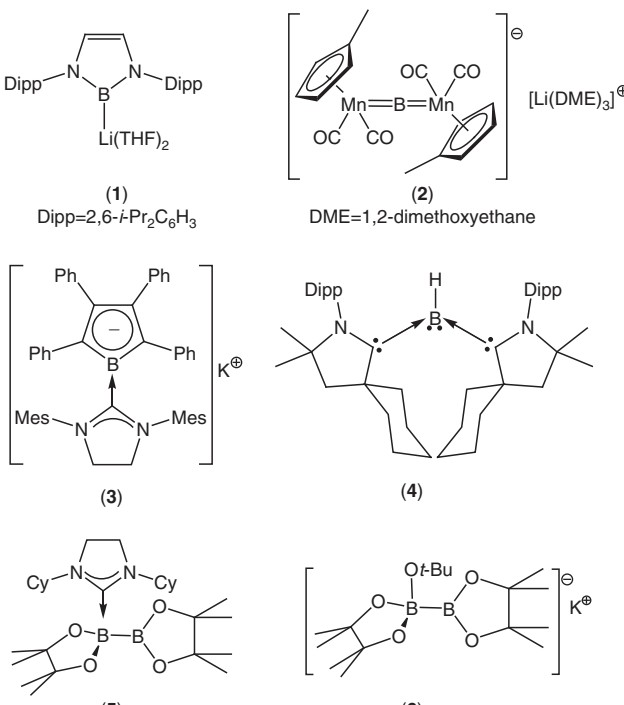

**Figure 1 | Selected examples of known molecules providing access to nucleophilic behaviour at boron.** The structures of compounds **1–6**.

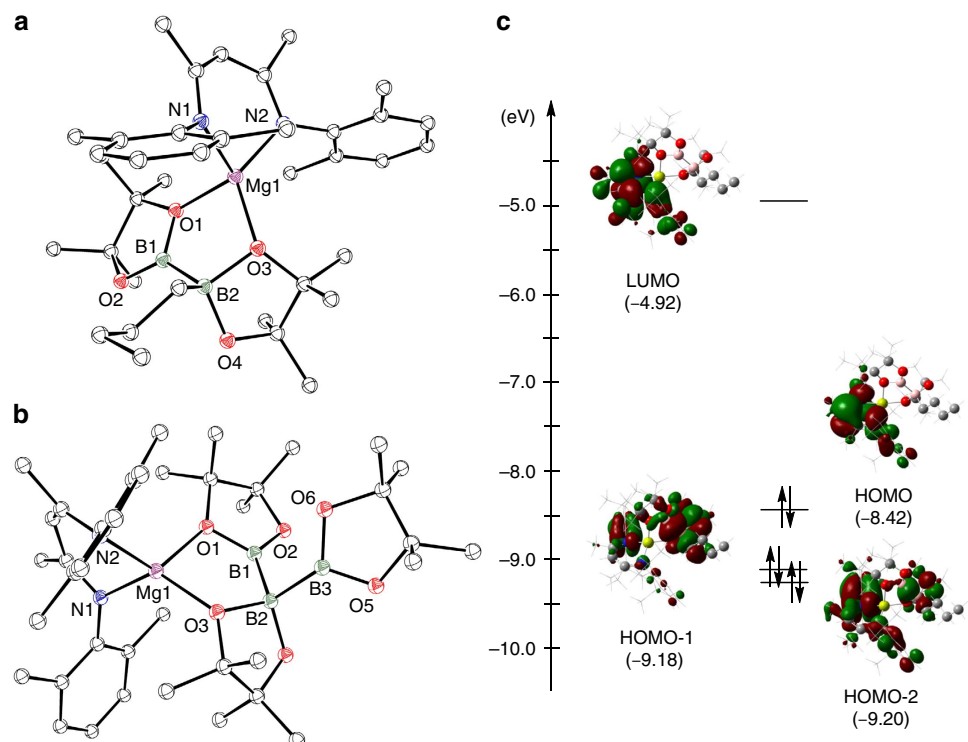

**Figure 2 | Synthesis of compounds 9–11.** Addition of one equivalent of $B_2pin_2$ to compound **7** provides compound **9**, while a similar reaction with two equivalents of $B_2pin_2$ provides compound **10**. Reaction of **9** or **10** with 4-dimethylaminopyridine provides facile access to the terminal magnesium boryl, compound **11**.

**Figure 3 | Single crystal X-ray structures of compounds 9 and 10 and the calculated frontier molecular orbitals of compound 9.** Thermal ellipsoid plots of (**a**) **9** and (**b**) **10** at 25% probability level; hydrogen atoms and iso-propyl methyl groups are omitted for clarity. Selected bond distances (Å) and angles (°): **9**, Mg(1)–O(1) 2.0768(8), Mg(1)–O(3) 1.9461(8), Mg(1)–N(1) 2.0688(9), Mg(1)–N(2) 2.0592(9), B(1)–B(2) 1.7503(18), C(42)–B(2) 1.6114(17), N(2)–Mg(1)–N(1) 97.28(4), O(3)–Mg(1)–O(1) 90.01(3). **10**. Mg(1)–O(1) 2.0563(17), Mg(1)–O(3) 1.9376(17), Mg(1)–N(1) 2.0515(19), Mg(1)–N(2) 2.0665(19), B(1)–B(2) 1.722(4), B(2)–B(3) 1.746(4), N(1)–Mg(1)–N(2) 95.21(8), O(3)–Mg(1)–O(1) 90.28(7), B(1)–B(2)–B(3) 100.3(2). (**c**) Natural bond orbitals and energies of the frontier molecular orbitals of **9**.

earlier compounds were prepared by the hydroboration of the B=B double bonds in stabilised diborenes, compound **10** may be rationalized as resulting from the displacement of $n$-BuBpin from compound **9** and the formal addition of a [Bpin]⁻ anion to bis(pinacolato)diboron.

DFT calculations performed on both compounds **9** and **10** provided optimized structures that corresponded closely with those obtained from the experimental X-ray diffraction data. Although the [pinB-Bpin($n$-Bu)]⁻ component of compound **9**

may be considered as broadly analogous to the anions of species such as **6** (ref. 44), the ordering of its key frontier orbitals militates against its straightforward action as a source of the [Bpin]⁻ anion. Figure 3c illustrates that the HOMO of **9** comprises the π-system of the β-diketiminate ligand while the B–B bond is represented by the HOMO-1 and HOMO-2, which are some 0.76 and 0.78 eV lower in energy, respectively. Although we have observed that this electronic structure results in a notable non-innocence of the β-diketiminate framework during reactions

of **9** with representative electrophiles, these observations will be described elsewhere.

The apparently facile displacement of *n*-BuBpin from the coordination sphere of **9** by the weakly basic (B$_2$pin$_2$) led us to speculate that treatment of **9** with a strongly coordinating base would effect a similar elimination of the alkyl borane to provide a terminal magnesium boryl. A reaction of compound **9** with a stoichiometric equivalent of 4-dimethylaminopyridine (DMAP) resulted in the complete consumption of **9** and the appearance of a set of resonances consistent with the production of a single new β-diketiminate environment in the resultant $^1$H NMR spectrum. The corresponding $^{11}$B NMR spectrum comprised two signals at δ 37.2 and − 5.40 p.p.m., which were assigned to *n*-BuBpin and the boron centre within a new compound **11**, respectively. While this latter chemical shift is to significantly higher field than the values observed for boron nuclei within derivatives of the anion in compound **1** (typically δ ∼ 35 p.p.m.)[7,9], unequivocal evidence of the identity of **11** was obtained from an X-ray diffraction analysis performed on colourless single crystals obtained from hexane solution. This experiment confirmed the constitution of **11** as a four-coordinate magnesium derivative (Fig. 4a) in which three of the magnesium to ligand contacts are provided by the nitrogen donors of the β-diketiminate ligand and a single unidentate DMAP ligand. The fourth coordination site of **11** is occupied by the [Bpin]$^-$ unit, which is ligated to magnesium through its *sp$^2$* boron centre as a terminal boryl ligand. The Mg(1)–B(1) distance [2.324(2) Å] lies within the range observed in three reported magnesium derivatives synthesized by reactions of compound **1** with MgBr$_2$ [2.281(6)–2.377(4) Å] and is comparable to the Mg–B bond length of a closely related and very recently reported *N*-heterocyclic 1,2,4,3-triazaborol-3-yl-magnesium species [2.341(7) Å], which was prepared by an analogous salt elimination route[14,27]. These data indicate that the Mg–B interaction within **11** possesses a similarly polarized nature. In contrast to these previously described species, all of which display a marginally distorted tetrahedral geometry at magnesium, the N(1)–Mg(1)–B(1) and N(2)–Mg(1)–B(1) bond angles [136.26(7)°; 115.90(7)°] subtended by B(1) and the nitrogen donors of the β-diketiminate ligand with magnesium are significantly more obtuse than the B(1)–Mg(1)–N(3) angle formed with the DMAP ligand [104.48(7)°]. As a result of these distortions, B(1) lies only 0.13 Å above the N(1)–C(2)–C(3)–C(4)–N(2) least squares plane such that B(1), N(1) and N(2) are effectively coplanar and form the basal plane of a distorted trigonal pyramid. Consequently, Mg(1) lies only 0.49 Å above the least squares plane defined by B(1), N(1) and N(2), with N(3) at the apex of the pyramid. The formation of compound **11** could

also be achieved through the addition of a single molar equivalent of DMAP to compound **10** and resulting in the irreversible displacement of B$_2$pin$_2$ (Scheme 2).

DFT calculations were carried out to interrogate the origins of the structure of **11**. The pyramidal geometry was very well replicated by the complete optimized structure as were the most relevant bond lengths across the molecule. In contrast to the ordering of the highest energy bonding molecular orbitals of **9**, the Mg–B σ-bonding interaction was found to be represented by the HOMO of **11**, which extends across the entirety of the β-diketiminate π-system and the effectively orthogonal pinacolate ligand (Fig. 4b). We suggest that this delocalization most likely provides a mechanism for charge dissipation which is intrinsic to the stability of the magnesium boryl unit. This observation is further borne out by examination of the calculated charge of the boron atom ( + 0.318), which is significantly more positive than that previously calculated for lithium species such as **1** (*vide infra*)[9]. In common with such *N*-heterocyclic boryls, and consistent with Schleyer's earlier theoretical predictions[13], we suggest that the singlet boryl anion in **11** is stabilised not only by its significantly covalent interaction with magnesium but also by the σ-inductive effects of the pinacolate oxygen atoms ( − 0.725, − 0.736) and the π-acceptor character of the β-diketiminate ligand system. This latter feature, in particular, presents future opportunities for the further tuning of the nucleophilic character of the [Bpin]$^-$ unit. The LUMO of **11**, meanwhile, is largely represented by the π* system of the coordinated DMAP ligand (Fig. 4b).

**Reactivity of compound 11 as a boron nucleophile.** Although the potential utility of species such as compound **1** lie in their ability to engage in reactions with organic electrophiles, reactivity studies of such lithium boryls have highlighted an ambiphilic character[9]. Reactions of **1** with organohalides, RX, have been shown to result in not only nucleophilic substitution at carbon to provide the desired alkyl borane but also halogen abstraction and resultant haloborane formation. DFT calculations on these systems have rationalized this reactivity as resulting from the operation of competitive thermodynamic (to provide the S$_N$2 product) and kinetic reaction pathways[50]. Furthermore, the more kinetically accessible halogen abstraction pathway was deduced to be promoted by organohalides bearing heavier halogens of lower electronegativity and a higher ability to engage in hypervalent bonding. With these observations in mind, we assayed the reaction of compound **11** with iodomethane, reasoning that this substrate would provide a high potential for competitive halogen

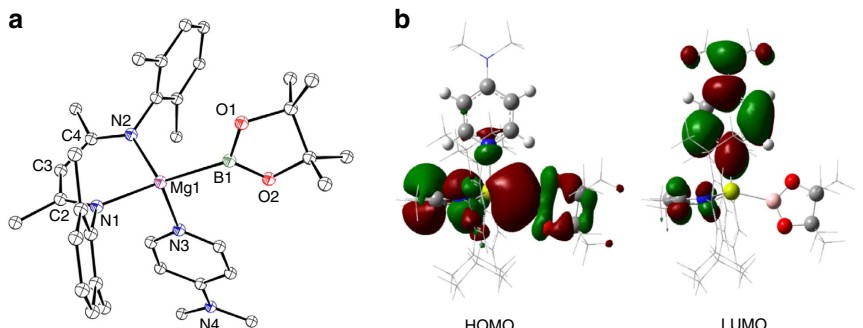

**Figure 4 | Single crystal X-ray structure of compound 11 and its calculated frontier molecular orbitals. (a)** Molecular structure of **11** as determined by X-ray crystallography. Thermal ellipsoids at 25% probability level; hydrogen atoms and iso-propyl methyl groups are omitted for clarity. Selected bond distances (Å) and angles (°): Mg(1)–N(1) 2.0798(14), Mg(1)–N(2) 2.0797(15), Mg(1)–N(3) 2.1308(16), Mg(1)–B(1) 2.324(2), N(1)–Mg(1)–N(3) 101.74(6), N(1)–Mg(1)–B(1) 136.26(7), N(2)–Mg(1)–N(1) 91.22(6), N(2)–Mg(1)–N(3) 102.70(6), N(2)–Mg(1)–B(1) 115.90(7), N(3)–Mg(1)–B(1) 104.48(7). **(b)** Natural bond orbitals for the frontier molecular orbitals of **11**.

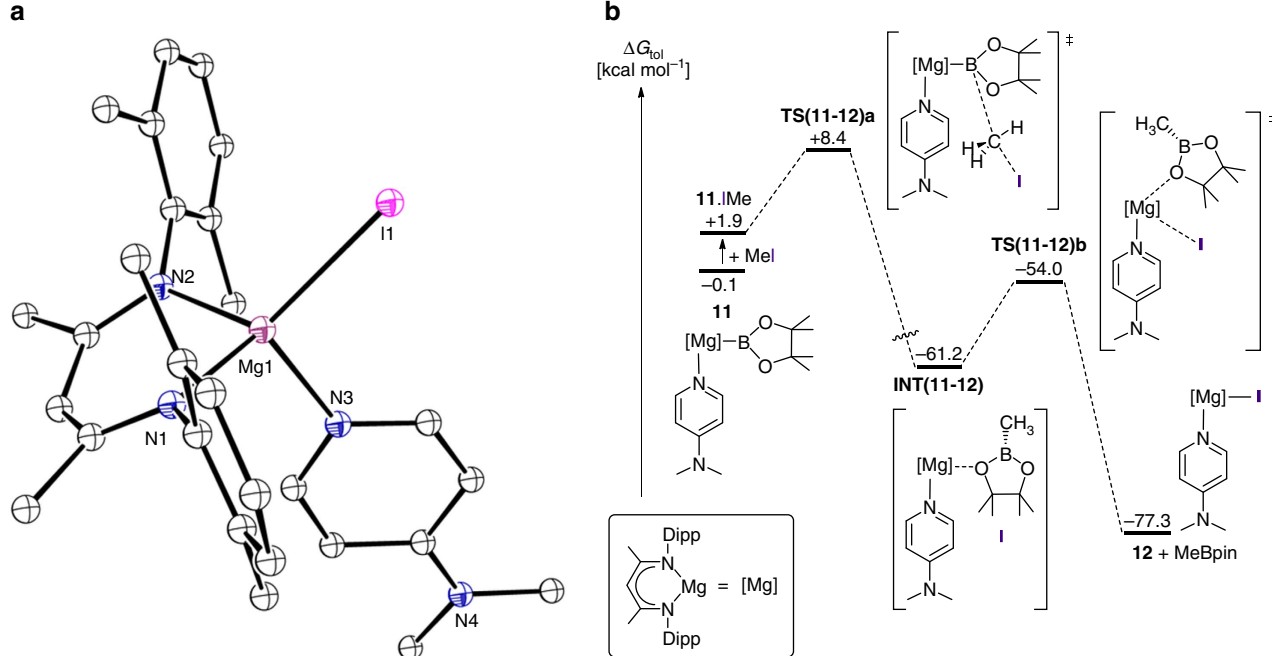

**Figure 5 | Single crystal X-ray structure of compound 12 and the calculated free energy profile for the reaction of 11 with MeI.** (**a**) Molecular structure of **12** as determined by X-ray crystallography. Thermal ellipsoids at 40% probability level; hydrogen atoms and iso-propyl methyl groups are omitted for clarity. Selected bond distances (Å) and angles (°): Mg(1)–N(1) 2.0481(15), Mg(1)–N(2) 2.0416(16), Mg(1)–N(3) 2.0887(16), Mg(1)–I(1) 2.6567(6), N(1)–Mg(1)–N(3) 107.60(7), N(1)–Mg(1)–I(1) 123.52(5), N(2)–Mg(1)–N(1) 93.70(6), N(2)–Mg(1)–N(3) 107.37(7), N(2)–Mg(1)–I(1) 118.84(5), N(3)–Mg(1)–I(1) 104.69(5). (**b**) DFT calculated free energy (kcal mol$^{-1}$) profile for the reaction of compound **11** with iodomethane (in toluene).

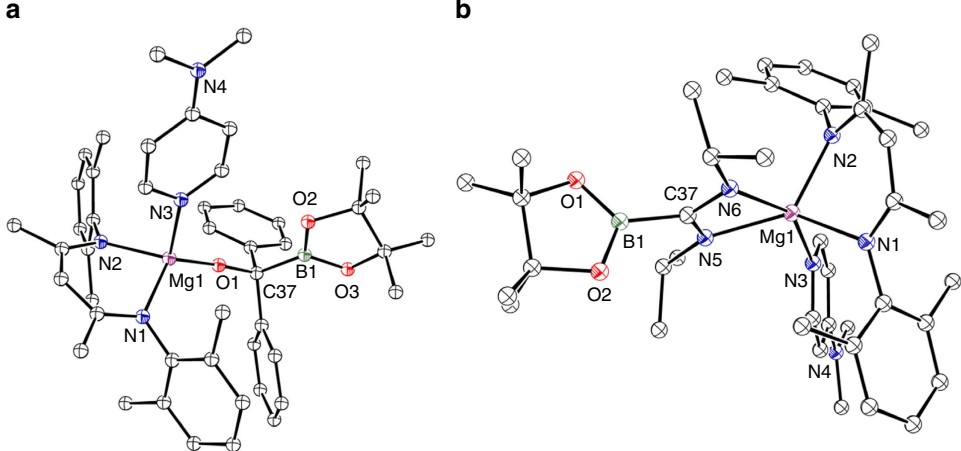

**Figure 6 | Single crystal X-ray structure of compounds 13 and 14.** Thermal ellipsoid plots of (**a**) **13** and (**b**) **14** at 25% probability level; hydrogen atoms and iso-propyl methyl groups of the Dipp substituents are omitted for clarity. Selected bond distances (Å) and angles (°): **13**, Mg(1)–O(1) 1.8182(11), Mg(1)–N(1) 2.0675(13), Mg(1)–N(2) 2.0787(13), Mg(1)–N(3) 2.1421(14), O(1)–C(37) 1.3946(18), C(37)–B(1) 1.594(2), O(1)–Mg(1)–N(1) 122.21(6), O(1)–Mg(1)–N(2) 122.17(5), O(1)–Mg(1)–N(3) 112.33(6), N(1)–Mg(1)–N(2) 91.41(5), N(1)–Mg(1)–N(3) 104.34(5), N(2)–Mg(1)–N(3) 100.48(5), C(37)–O(1)–Mg(1) 175.26(11), O(1)–C(37)–B(1) 110.56(13). **14**, N(1)–Mg(1) 2.1079(10), N(2)–Mg(1) 2.1008(10), N(3)–Mg(1) 2.2083(10), N(5)–Mg(1) 2.1100(10), N(6)–Mg(1) 2.2124(10), C(37)–B(1) 1.5972(17), N(1)–Mg(1)–N(1) 90.79(4), N(5)–Mg(1)–N(6) 62.73(4), N(5)–C(37)–N(5) 115.58(10).

abstraction. A reaction performed in $d_8$-toluene and monitored by [1]H NMR spectroscopy provided a single new β-diketiminate compound (**12**) and evidenced the production of MeBpin which was observed as two singlet resonances at δ 0.32 (B-*Me*) and 1.01 p.p.m. (C-*Me*) within the first point of analysis. The corresponding [11]B NMR spectrum comprised a single resonance at δ 36.9 p.p.m., which was also assigned to the formation of the product of nucleophilic iodide displacement, MeBpin (ref. 51). The constitution of the magnesium-containing side product (**12**) was confirmed through a further single crystal X-ray diffraction analysis as the anticipated β-diketiminato

magnesium iodide (Fig. 5a), which preserves a monomeric constitution through the retention of the coordinated DMAP ligand. Although we cannot discount the possibility that the mechanism of boron methylation takes place via a radical-based pathway[52], it is notable that a further reaction of compound **11** and MeI performed in the presence of the potential radical trap 9,10-dihydroanthracene provided identical conversion to compound **12** and MeBpin and evidenced no consumption of the aromatic hydrocarbon (Supplementary Fig. 14). DFT analysis of the reaction pathway (Fig. 5b) between **11** and MeI also confirmed the facility of the nucleophilic iodide displacement.

The overall reaction was found to be highly exergonic ($\Delta G_{tol} = -77.3 \, \text{kcal mol}^{-1}$) with a reaction barrier of only $8.5 \, \text{kcal mol}^{-1}$ presented by the $S_N2$ transition state (**TS(11–12)a**; $\Delta G_{tol} = +8.4 \, \text{kcal mol}^{-1}$) in which the boryl anion acts as an unambiguous nucleophile through a classical backside attack on the iodomethane carbon atom. This process provides an intermediate (**INT(11–12)**; $\Delta G_{tol} = -61.2 \, \text{kcal mol}^{-1}$) in which the Me-Bpin is coordinated to the Mg complex through one of its oxygen atoms ($Mg \cdots O = 2.11 \, \text{Å}$) and the iodide persists within the outer coordination sphere of the complex ($Mg \cdots I = 5.8 \, \text{Å}$, see Supplementary Fig. 2). Concerted coordination of iodide to magnesium and dissociation of the methyl borane product occurs with a barrier of $7.2 \, \text{kcal mol}^{-1}$, (**TS(11–12)b**; $\Delta G_{tol} = -54.0 \, \text{kcal mol}^{-1}$), to form the ultimate product **12** ($\Delta G_{tol} = -77.3 \, \text{kcal mol}^{-1}$).

The nucleophilic potential of the $[Bpin]^-$ anion of compound **11** was further confirmed through its reaction with the representative non-halogenated organic electrophiles, benzophenone and $N,N'$-di-isopropyl carbodiimide. In both cases, reactions performed in toluene solution resulted in the immediate formation of single new products. The structure of compound **13**, from the reaction with benzophenone, was deduced through an X-ray diffraction analysis (Fig. 6a), which confirmed the nucleophilic addition of the $[Bpin]^-$ to the carbonyl carbon to produce an unprecedented diphenyl(boryl)alkoxide species that also retains the DMAP ligand coordinated to magnesium. Although this behaviour is somewhat analogous to the reported reactivity for s-block derivatives of the anion of compound **1** with benzaldehyde[9,53], the high specificity of the current reaction is particularly notable. Similarly, the reaction of **11** with $N,N'$-di-isopropyl carbodiimide provided clean insertion with C–B bond formation and the resultant production of a magnesium bora-amidinate derivative (**14**, Fig. 6b). Although the formation of this new anion is completely analogous to the widely employed reaction of carbodiimides with the M–C bonds of polar σ-organometallics[54,55], the only precedent for similar M–B insertion is provided by a scandium derivative, which was itself prepared from compound **1** (ref. 18).

In conclusion, we have shown that an easily synthesized magnesium organometallic derivative may be readily converted by a straightforward two step reaction to a terminal magnesium boryl derivative, which behaves as a source of the widely used [Bpin] moiety with pronounced nucleophilic character. Although the isolation of compound **11** is dependent upon the kinetic and electronic stability provided by a bulky β-diketiminate ligand, these observations indicate that the future chemistry of boryl anions themselves need not be constrained by a need for extreme steric protection and unattractive reducing conditions. Furthermore, we suggest that these primary results could facilitate an entirely new field through which the preparation of previously inaccessible organoboron molecules may be enabled.

## Methods

For synthetic details and analytical data for compounds **9**–**14** and details of the DFT calculations contained in this paper see Supplementary Methods. The computed relative energies (kcal mol$^{-1}$) for the reactions of compounds **9** and **11** are compiled in Supplementary Table 1. Cartesian coordinates for all the DFT-computed geometries are provided in Supplementary Data 1. DFT-computed geometries for the addition of MeI to complex **11** are shown in Supplementary Fig. 2. For a view of the single crystal X-ray structure resulting from the co-crystal of compounds **9** and **10** see Supplementary Fig. 3. For details of the single crystal X-ray diffraction analysis of compounds **9**–**14** see Supplementary Methods and Supplementary Table 2. For $^1H$, $^{11}B\{^1H\}$ and $^{13}C\{^1H\}$ NMR spectra of the compounds in this article, see Supplementary Figs 4–21.

**Data availability.** X-ray crystallographic data for compounds **9**–**14** plus data pertaining to a co-crystal of **9** and **10** are freely available from the Cambridge Crystallographic Data Centre (CCDC 1511682 to CCDC 1511688). All other data are available from the authors upon reasonable request.

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

## Acknowledgements

This work was supported by the Engineering and Physical Sciences Research Council (EP/I014519/1 and EP/N014456/1). We also thank the Australian Government Endeavour Fellowship Programme (A.L.C.).

## Author contributions

C.W. and A.L.C. carried out the initial syntheses of compounds **9–11** respectively. A.-F.P. carried out the synthesis and characterization of all the other compounds. M.F.M. collected the single crystal X-ray crystallographic data and solved the crystal structures. C.L.M. devised, carried out and interpreted the DFT calculations. M.S.H. generated and managed the project and wrote the manuscript. All the authors discussed the results and commented on the manuscript.

## Additional information

**Competing interests:** The authors declare no competing financial interests.

**Publisher's note**: 

