## [Peer review file · Nature Communications]

Reviewers' comments:

Reviewer #1 (Remarks to the Author):

The submitted article by McMullin, Hill, and colleagues introduces the convenient two steps formation of a stable magnesium complex with a boryl group. Nucleophilic nature of the title compound has been demonstrated by the reactions with MeI, ketone, as well as carbodiimide. The work technically appears to be sound, as all newly synthesized molecules are for the most part sufficiently characterized. Computational approach supports experimental observation. The article is straightforward, understandable and presented well.

In the reported studies (references 34-35), generation of stable anionic adducts of B2pin2 with Lewis bases and its behaviour as boron-nucleophile have already been described. Analogous formation of compound 9 from butyl magnesium complex 7 with Bpin2 is thus not surprising. Additionally, since Yamashita and Nozaki reported the reaction of borylmagnesium with benzaldehyde (reference 13), the last part of this article which is unequivocally related to those results by Yamashita, shows only incremental progress on the nucleophilic reactivity of Mg-B bonded compound.

Meanwhile, although the entire process for formation of Mg-B from Mg-nBu (7) and B2pins, explicitly mirrors the authors' previous work (ref 36) about formation of Mg-Si from Mg-nBu (7) and Si-Bpin, one novelty seen in present study is the formation of the Mg-B bond (11) from compound 9 by addition of DMAP. Because extant protocols to generate stable boryl metals are limited to the reduction of the corresponding halo-boranes or deprotonation of the B-H precursor (ref 32), the authors finding of the way to form Mg-B bond is fundamentally sound. This particular work should lead to more interesting results from those looking to the scope of this Mg-B bond formation using other diboron substrates such as catB-Bcat. Unfortunately, the current work employs only B2pin2, which is still not sufficient enough to attract many synthetic chemists since generation of Bpin nucleophile from B2pin2 and its application even in catalysis have been already well documented (ref 35).

I do not see the yield of compound 11 in both main text and even the Supplementary materials, and therefore cannot confirm the efficiency of this unique reaction.

Although the part of the DMAP-induced formation of boron-magnesium bond is sound, overall present results seem to be premature somehow, and the article can be significantly improved by addressing the points described below:

- Diboron substrates other than B2pin2 should be tested to demonstrate that the authors' strategy has an advantage over the other base-B2pin2 adduct system (ref 34-35). Moreover, significance of terminal Mg-B bonded compound 11 rather than the 7-B2pin2 adduct (9) in terms of the reactivity, has to be proved. In the present article, however, no support at all is evident for the necessity to pre-form the Mg-B bonded compound (11) for the formation of the compounds 12, 13, 14. Did authors examined if the reactions of compound 9 itself (as well as 10) with MeI, benzophenone, and carbodiimide could afford respective products 12,13,14, or not? Additionally, it is curious to check if compounds 10 and 11 are interchangeable by addition of DMAP to 10 or B2pin2 to 11.

- “current candidate derivatives are limited by a need for bulky substituents and highly reducing conditions”. This statement is not accurate. Transient boryl anions with small substituents, for instance, BH₂ anion supported by NHC (ACIE, doi: 10.1002/anie.201004215), was reported. Compounds that authors point out here must be only those of isolable species, which should be explicitly alluded to.
- “These observations indicate that the future chemistry of boryl anions need not be constrained by a need for extreme kinetic protection and unattractive reducing conditions”. This statement sounds somehow misleading because the authors’ magnesium complex has in fact a very bulky beta-diketiminato ligand which of course kinetically stabilizes the Mg-B moiety in 11. In contrast to the reported isolable boryl metals supported bulky substituents on the boron, the authors use kinetic protection from the ligand on the Mg allowing for the isolation of compound 11, which should be explicitly alluded to.
- As a general review, the first chapter in the latest book "Synthesis and Application of Organoboron Compounds, Fernández, E.; Whiting, A., Eds. Springer International Publishing" can also be cited. Additional references “Chem Commun doi: 10.1039/C4CC03497J” and “ACIE, doi: 10.1002/anie.201509289, doi: 10.1002/anie.201405201, doi: 10.1002/anie.201504579” can also be cited in the introductory section.
- On page 1. Not ‘in 2007’ but ‘in 2006’ (see ref 7).
- In the Supplementary material, the authors write “the yield of 9 is 100%”. As the compound 7 (50 mg) and B2pin2 (25.4 mg) were utilized, the product 9 (75.9 mg) is scientifically inappropriate.

Reviewer #2 (Remarks to the Author):

The provision of genuine boron centered nucleophiles has significantly enriched organic and organometallic syntheses in recent years and has made a number of highly unusual boron-element combinations available, which were thought to be non-existing. However, the number of strong boron nucleophiles is very much restricted to a handful of examples and their synthesis is often tedious.

McMullin, Hill and coworkers present an elegant protocol, starting from readily accessible or commercially available precursors, which provides the title compound as another boron centered nucleophile. Its somewhat unusual generation as well as its electronic groundstate have been sufficiently rationalized by experimental and computational studies and its nucleophilic character has been proven by a number of trapping reactions.

I find the reported straightforward synthesis highly appealing and I agree with the authors that their finding most likely will have a major impact on the contemporary use of boron in organometallic, organic and inorganic synthesis. Thus, I suggest publication of the present study in Nature Communications after some minor revision.

- In the introduction the authors refer to representative boron nucleophiles. One might add information on particular diborenes, which are known to be exceptionally electron rich (e.g. Angew. Chem. Int. Ed. 2015, 54, 359) and act as (neutral!) nucleophiles (Angew. Chem. Int. Ed. 2016, 55, 5606). Besides, the highly nucleophilic dianion (!!) B(CN)₂²⁻ should not be omitted (see e.g. Angew. Chem. Int. Ed. 2015, 54, 11259 and in particular Angew. Chem. Int. Ed. 2011, 50, 12085).

- The authors use MeI as one substrate to prove the nucleophilic character of the title compound. While this is done quite often, it should be mentioned that MeI in particular is prone to SET processes, which might mimic nucleophilic substitutions but are in fact radical reactions (see e.g. Angew. Chem. Int. Ed. 2014, 53, 5453 and references therein). While I agree with the author's view that the calculated mechanism in Fig. 5 supports the reactivity of 11 as a nucleophile here, I would welcome some critical remarks on the careful interpretation of the the MeI reactivity here along with some more experimental evidence. Possibly 11 would also react with a less ambiguous substrate such as n-BuCl, or the reaction with MeI could be repeated in the presence of a radical trap.

Reviewer #3 (Remarks to the Author):

As an X-ray crystallographer I only feel comfortable commenting on the validity of the structures presented in the manuscript. The checkCIF reports suggest no particular issues except the Alert A in the co-crystal of 9 and 10. In this particular case the Alert derives from the absence of any information from SQUEEZE in the CIF. SQUEEZE is mentioned in the supplementary information, however none of the output is supplied in the CIF itself. The authors suggest two molecules of pentane were removed from the lattice, and while there's no reason to suspect otherwise, it is important to include the SQUEEZE output in the CIF. I would also suggest adding an appropriate reference to the SQUEEZE program

Compounds 10, 11, 12, 13 and the 9+10 co-crystal all exhibit disorder of some sort. The disorder is discussed in the SI, however no details of the steps taken to properly model any disorder are evident in the text or the CIFs. The version of ShelXL employed by the authors should (in conjunction with Olex2) automatically include the final .res file. This should be standard with any submitted structure in the absence of any detailed discussion of the refinement process. The disorders were modelled with absolute site occupancies, rather than by refining the site occupancies. No reason is given for this, and no explanation was given as to how they came to use the values they did.

All these comments are not to suggest that any conclusions drawn from the structures are invalid, much of the disorder pertains to lattice solvent unrelated to the structure of interest, or of the orientation of the pinacol carbons. There is nothing to suggest these structures have been improperly characterised.

Referee 1

This referee was in favour of publication but requested the inclusion of significant additional material that would take the manuscript way beyond the scope of a high impact communication which is limited to a 5000 word maximum length.

Our submission describes a simple route to a synthetically useful boryl anion and provides a preliminary demonstration of its potential as a nucleophilic reagent. The referee feels that our study of B_2pin_2 is, however, 'not sufficient to attract many synthetic chemists'. We disagree completely with this opinion and are of course in the process of extending this reactivity to alternative diborane substrates. This additional research, however, is not yet as mature as our study of the pinacolate derivative, which was selected for initial study precisely because of its widespread application as a commercially available diboron reagent in the chemical literature. Although our limited preliminary examination of other commercially available diboranes appears to show that this reactivity may well be generalised, we feel these extensions, along with any divergence in their chemical behaviour, will be best considered in more extensive subsequent publications and not the current communication.

Referee 1 also requests that we divulge whether reactions of compound 9 with MeI, benzophenone and carbodiimide similarly provide compounds 12, 13 and 14. This is impossible as these latter compounds also contain an equivalent of coordinated DMAP, which would not be included in any of these reactions. This observation aside, and as we allude to on page 5 of our submission, the structure of compound 9 results in some quite significant divergence in its reactivity with these substrates. The inclusion of this further chemistry (including up to five extra crystallographically characterised compounds and an associated computational study) would necessitate appropriately detailed discussion which would, again, significantly elongate the manuscript and dilute the overall emphasis of the communication on the generation and rational reactivity of a true Mg boryl.

The yield of compound 11 is now included in the experimental section of the revised Supplementary Information.

As the referee suggests, compound 10 may be irreversibly converted to compound 11 through addition of DMAP. This is now illustrated in Figure 2, described in the figure legend and is described prior to Figure 4 on page 6 through the addition of the following sentence; ‘The formation of compound 11 could also be achieved through the addition of a single molar equivalent of DMAP to compound 10 and resulting in the irreversible displacement of B₂pin₂ (Scheme 2).’

The generation of the BH₂ anion is now described in the introduction of the paper and is cited as reference 42. Similarly, the other relevant references provided by the referee are included as additional references 12 (review), 36 – 38 and 41. Reference to these papers is now included in the introduction to the paper through the incorporation of the following modified text on page 2 of the manuscript; ‘Notable exceptions are Braunschweig’s dimetalloborylene and borolyl anions (e.g. 2 and 3)^{32,33} Bertrand and Kinjo’s carbene-stabilised monovalent boron species (e.g. 4)³⁴⁻³⁸ and a remarkable tricyanoborandiyl dianion.³⁹⁻⁴¹ Although the transient generation of {BH₂}⁻ anion equivalents has also been achieved within the coordination sphere of 1,3-bis-(2,6-diisopropyl-phenyl)imidazol-2-ylidene,⁴² the syntheses of all these species require strongly reducing and potentially problematic reaction conditions (e.g. Li metal or C₈K) to achieve the formal B(I) oxidation state. The successful isolation of these compounds is also typically dependent on the high degree of kinetic stabilisation provided by sterically demanding substituents directly about the boron centre.’

Referee 1 appears to consider our concluding comments, with regard to the need for kinetic protection in the isolation of the boryl species, as somewhat misleading or disingenuous. This was certainly not our intention as were referring specifically to the Bpin unit itself. We fully acknowledge that the isolation of compound 11 is profoundly dependent on the kinetic stability provided by the beta-diketimate ligand and this feature of the chemistry is now clearly ascribed through the following modification to the concluding statement on page 9; ‘Although the isolation of compound 11 is dependent upon the kinetic and electronic stability provided by a bulky β-diketimate ligand, these observations indicate that the future chemistry of boryl anions themselves need not be constrained by a need for extreme steric protection and unattractive reducing conditions.’

The year of publication of Yamashita’s publication of compound 1 has been corrected to 2006 on page 1.

The stated yield of compound 9 was a typographical error. This has been corrected to 75.4 mg in the revised Supplementary Information.

Referee 2

This referee was strongly supportive of publication after minor prescribed revision.

The additional citations provided are now included in the introduction to the communication as references 30, 31, 39 and 40 (also suggested by referee 1) and are cited in the modified text highlighted in our response to referee 1 above.

The referee quite rationally suggests that the reaction of compound **11** with MeI could also ensue through a radical based pathway. We are already in the process of carrying out a complete assessment of the reactivity of the boryl species with a range of additional alkyl halides. Our observations thus far indicate that the mode of reactivity with MeI appears to be completely general. As was the case for the inclusion of the additional reactivity suggested by Referee 1, we feel that incorporation of these extra results would unnecessarily lengthen our submission without adding any meaningful insight. We do, however, fully acknowledge that a radical-based process is an alternative pathway worthy of consideration and, as suggested, have now carried out the reaction in the presence of the radical trap, 9,10-dihydroanthracene. Although this modification to the reaction conditions does not induce any appreciable change to the course or the specificity of the reaction we feel it is prudent to highlight that we cannot yet completely exclude the possibility that the reactivity may involve some radical-derived character. We, thus, include the following additional comment on page 7 of the revised manuscript; ‘Although we cannot discount the possibility that the mechanism of boron methylation takes place via a radical-based pathway,⁵² it is notable that a further reaction of compound 11 and MeI performed in the presence of the potential radical trap 9,10-dihydroanthracene provided identical conversion to compound 12 and MeBpin and evidenced no consumption of the aromatic hydrocarbon (Figure S14).’ The additional citation to the work of Braunschweig and co-workers is now incorporated as reference 52. The in situ NMR spectrum resulting from this reaction is also now included as corroborative content as Figure S14 in the revised Supplementary Information and the subsequent figures have been renumbered accordingly.

Referee 3

This referee made no comment on the new chemistry described in the paper but made some minor criticism of the presentation of the crystallographic data included in our submission.

We have redeposited the larger forms of the CIFs that now include the corresponding RES and HKL files in each case for all of the structures in the paper. Consequently, as requested by the referee, the SQUEEZE details are now evident for the co-crystal in the SI.

In accord with the referee’s request, the narrative with regard to the disorder is now included universally in the redeposited CIF files.

REVIEWERS' COMMENTS:

Reviewer #2 (Remarks to the Author):

All concerns have been sufficiently addressed.
I suggest publication as is.